# Developing a Tanshinone IIA Memetic by Targeting MIOS to Regulate mTORC1 and Autophagy in Glioblastoma

**DOI:** 10.3390/ijms25126586

**Published:** 2024-06-14

**Authors:** Sonia Shinhmar, Judith Schaf, Katie Lloyd Jones, Olivier E. Pardo, Philip Beesley, Robin S. B. Williams

**Affiliations:** 1Centre for Biomedical Sciences, School of Biological Sciences, Royal Holloway University of London, Egham TW20 0EX, UK; sonia.shinhmar.2020@live.rhul.ac.uk (S.S.); judith.schaf@gmx.de (J.S.); katie.lloyd-jones@rhul.ac.uk (K.L.J.); p.beesley@rhul.ac.uk (P.B.); 2Division of Cancer, Department of Surgery and Cancer, Imperial College London, London SW7 2AZ, UK; o.pardo@imperial.ac.uk

**Keywords:** autophagy, cancer, *Dictyostelium discoideum*, drug discovery, GATOR2, glioblastoma, MIOS, mTORC1, sestrin, tanshinone IIA

## Abstract

Tanshinone IIA (T2A) is a bioactive compound that provides promise in the treatment of glioblastoma multiforme (GBM), with a range of molecular mechanisms including the inhibition of the mechanistic target of rapamycin complex 1 (mTORC1) and the induction of autophagy. Recently, T2A has been demonstrated to function through sestrin 2 (SESN) to inhibit mTORC1 activity, but its possible impact on autophagy through this pathway has not been investigated. Here, the model system *Dictyostelium discoideum* and GBM cell lines were employed to investigate the cellular role of T2A in regulating SESN to inhibit mTORC1 and activate autophagy through a GATOR2 component MIOS. In *D. discoideum*, T2A treatment induced autophagy and inhibited mTORC1 activity, with both effects lost upon the ablation of SESN (sesn^-^) or MIOS (mios^-^). We further investigated the targeting of MIOS to reproduce this effect of T2A, where computational analysis identified 25 novel compounds predicted to strongly bind the human MIOS protein, with one compound (MIOS inhibitor 3; Mi3) reducing cell proliferation in two GBM cells. Furthermore, Mi3 specificity was demonstrated through the loss of potency in the *D. discoideum* mios^-^ cells regarding cell proliferation and the induction of autophagy. In GBM cells, Mi3 treatment also reduced mTORC1 activity and induced autophagy. Thus, a potential T2A mimetic showing the inhibition of mTORC1 and induction of autophagy in GBM cells was identified.

## 1. Introduction

Tanshinone IIA (T2A) is a bioactive compound derived from the root extract of Danshen (*Salvia miltiorrhiza* Bunge; family Lamiaceae) [1]. T2A has been demonstrated to function as a cancer treatment through a range of pathways including the AMPK (5′ adenosine monophosphate-activated protein kinase), MAPK (mitogen-activated protein kinase), and PI3K (phosphatidylinositol 3-kinase) signalling pathways [2], and many studies have defined a mechanism through the targeted inhibition of mTORC1 (mechanistic target of rapamycin complex 1) [3,4,5,6,7,8]. The mTORC1 pathway is a key metabolic regulator in many cell types, where the activity of this complex determines cellular homeostasis and a range of processes including cell proliferation and autophagy [9,10]. T2A has also been demonstrated to induce autophagy in a range of cancer models [8,11,12]. Thus, clarity regarding this mechanism may provide improved treatment approaches.

Autophagy is a metabolic process by which cells recycle components such as damaged proteins in specialised vesicles called autophagosomes. Although initially considered to function during starvation, it has been widely implicated in a range of diseases [13]. The role of autophagy in cancer development and progression is determined by tumour stage, location and environment [14], and autophagy provides a means of reducing cancer cell propagation, metastasis and further mutagenesis [15,16,17,18]. T2A has recently been shown to inhibit mTORC1 activity through the increased expression of sestrin 2 (SESN), both in the model system *D. discoideum* and in glioblastoma multiforme (GBM) cells [19]. The increased activity of SESN2 has also been shown to increase autophagy by inhibiting mTORC1 and additionally playing a role in mitophagy and the regulation of inflammation [20,21]. Thus, T2A may increase autophagy in GBM cells through the upregulation of SESN activity. However, the efficacy of T2A in reducing mTORC1 activity in GBM may be dependent upon p53 activity [19], and since p53 activity is commonly lost in cancers [22], alternate approaches to mimic the effect of T2A to induce autophagy independent of p53 need to be examined.

*D. discoideum* has been widely used as a tractable model to investigate molecular mechanisms for bioactive compounds as disease treatments [23]. Compounds investigated as disease treatments using *D. discoideum* include medium-chain fatty acids and cannabinoids in epilepsy, bipolar disorder, and multiple sclerosis treatment [24,25,26,27,28] and polyphenols in polycystic kidney disease and Alzheimer’s disease treatment [29,30]. The advantage of this model in molecular science research is a low redundancy genome with numerous conserved signalling pathways and orthologous proteins associated with human diseases [31], as well as the ability to rapidly ablate target genes to investigate a loss of potency towards bioactive compounds. The theory here is that the genetic ablation of a protein in *D. discoideum* alters cell signalling so the mutant can survive and proliferate, such that these cells show reduced sensitivity to bioactive compounds that target the deleted protein. Numerous studies have investigated medicinal treatments in *D. discoideum*-targeting mTORC1 activity, with results translated to primary mammalian neurons or human-derived cells [19,24,27]. Thus, *D. discoideum* provides an excellent model for drug development and translation.

The main aim of this study was to investigate the mechanism of action of T2A in the induction of autophagy and then the potential of developing T2A mimetics through targeting downstream proteins. The tractable model *D. discoideum* was employed since it was used to identify the T2A mechanism in regulating SESN expression to control mTORC1 activity [19]. Initially, the investigation focused on a role for T2A in inducing autophagy in this model, which was dependent upon SESN function. The molecular pathway involved in this induction was investigated by targeting a component of the GATOR2 (GAP Activity TOwards Rags 2) complex, MIOS (Meiosis Regulator For Oocyte Development), and the assessment of reduced T2A potency in regulating cell proliferation, mTORC1 activity, and autophagy induction. Then, an in silico screen of novel small molecule compounds was used to identify novel MIOS inhibitors, and the resulting compounds were investigated to reproduce the effect of T2A on the inhibition of GBM cell proliferation. From this analysis, shortlisted compounds were investigated for specificity using the *D. discoideum* mios^-^ mutant, and efficacy in GBM cell lines was analysed though the inhibition of mTORC1 activity and autophagy induction.

## 2. Results

### 2.1. T2A Treatment Increases Autophagy in D. discoideum

To validate the use of *D. discoideum* to investigate the T2A-dependent induction of autophagy, wild-type (Ax3) cells were transformed with an ATG8-GFP-tagged vector [32]. ATG8 (autophagy-related protein 8) is a key component of autophagosome production, and the fluorescently tagged protein allowed for the visualisation and analysis of cellular autophagosome numbers (Figure 1A). Initially, cells were exposed to starvation conditions (4 h in phosphate buffer), autophagosomes were imaged using fluorescent microscopy (Figure 1B), and autophagosome numbers were quantified and normalised to the cell size. Starvation significantly increased autophagosome formation from 0.39 per μm^3^ under solvent-only conditions (DMSO) to 0.89 autophagosomes per μm^3^ cell, providing a 128% increase (*p* < 0.0001) that was consistent with previous studies using C6 glioma stem-like cells [33]. The treatment of cells with T2A (12 μM, 24 h) at a concentration that inhibited mTORC1 [19] also significantly increased the number of autophagosomes from 0.30 to 0.57 per μm^3^, providing a 90% increase (*p* < 0.0001) (Figure 1B). Thus, autophagosome formation significantly increased upon T2A treatment in wild-type *D. discoideum* cells.

### 2.2. SESN Activity Is Required for T2A-Induced Autophagy Induction

Since the sesn^-^ mutant was previously used to identify the effect of T2A in *D. discoideum* [19], this mutant was used to investigate a role for this protein in T2A-dependent autophagy induction. Here, sesn^-^ cells were also transformed with the ATG8-GFP vector, and autophagy induction was monitored under starvation conditions and with T2A treatment. Under starvation conditions (4 h in phosphate buffer), autophagosome formation significantly increased from 0.43 to 0.94 autophagosomes per µm^3^ cell, representing an 118% increase (*p* < 0.0001) (Figure 1C) and indicating autophagosome formation was possible in cells lacking SESN. However, in sesn^-^ cells, T2A treatment (12 µM, 24 h) decreased the autophagosome number from 0.40 to 0.28 autophagosomes per µm^3^ cell, representing a 30% decrease (*p* < 0.0001) (Figure 1C). These results show that while the sesn^-^ mutant was able to form autophagosomes, it failed to form these in response to T2A treatment, indicating a requirement for SESN in T2A-induced autophagy.

### 2.3. MIOS Activity Is Required for T2A-Dependent Autophagy Induction

To overcome a potential need for the p53 activity of SESN in T2A autophagy induction [19,22], the MIOS protein was selected for analysis as a key component of the GATOR2 complex [34] in the SESN to mTORC1 pathway (Figure 2A). GATOR2 is composed of five types of proteins, with four MIOS subunits, two WDR24 subunits, two WDR59 subunits (WD repeat-containing proteins) and the accessory proteins SEH1L and SEC13 [35,36] (Figure 2B). MIOS was chosen as the most abundant protein in the complex, and the *mios* gene was ablated in *D. discoideum* (DDB_G0276673) using a CRISPR-Cas9 system [37] (Appendix A).

The *D. discoideum* mios^-^ mutant was initially used to investigate a role for MIOS in the T2A-dependent reduction in cell proliferation. In these experiments, wild-type (Ax3) and mios^-^ cells were exposed to a range of T2A concentrations (0–25 µM for 7 days) and cell numbers were recorded (Appendix A). This analysis showed that cell proliferation was decreased in wild-type cells with increasing T2A concentration (Figure 2C), and the effect was reduced in mios^-^ cells. The quantification of this analysis showed an IC_50_ of 1.7 µM in wild-type cells that increased to 13.3 µM in the mios^-^ mutant, representing an increase of 682% following the loss of MIOS. These data confirm a key role of MIOS (and hence the GATOR2 complex) in the effect of T2A on reducing cell proliferation in *D. discoideum*.

Next, the effect of T2A on mTORC1 activity and autophagy induction was investigated through SESN and MIOS [38] or via an independent pathway [39]. In these experiments, wild-type (Ax3) and mios^-^ cells were treated with T2A (25 µM for 24 h) and mTORC1 activity was assessed by the phosphorylation of 4EBP1 (eukaryotic translation initiation factor 4E (eIF4E)-binding protein 1) [19]. The treatment of wild-type cells with T2A reduced p-4EBP1 levels by 20% (*p* < 0.0062) (Figure 2D), and this effect was lost in *mios^-^* cells. To investigate a role for MIOS in T2A-dependent autophagy induction, autophagosome formation was also measured in mios^-^ cells (Figure 2E), In these experiments, the autophagosome number per µm^3^ cell significantly increased under starvation conditions from 0.43 to 0.74 (*p* < 0.0001), representing a 72% increase and evidencing active autophagy in this mutant. However, T2A treatment (12 µM, 24 h) caused a decrease in autophagosome production from 0.56 autophagosomes per µm^3^ cell to 0.45, representing a 20% reduction (*p* < 0.0004). Therefore, the presence of MIOS in the GATOR2 complex is necessary for the T2A-dependent inhibition of mTORC1 and the induction of autophagy in *D. discoideum*. 

### 2.4. The Identification of T2A Mimetics by Direct Binding to MIOS

To overcome a potential role for the loss of p53 in cancer impacting *sesn* expression and hence T2A activity, novel small molecules were investigated to target MIOS (Figure 3) [36]. Initially, an in silico screen was employed to analyse the potential binding of 4 million compounds against the active site β-propeller of the human MIOS protein (NP_001357009.1) (Figure 3A, Appendix A). From this analysis, 50 compounds providing the strongest predicted binding were individually docked against human MIOS to determine the Gibbs free energy (∆G) values. Interestingly, all compounds were predicted to bind to the inner side of the β-propeller structure of MIOS (Appendix A) except for Mi3 and Mi1, which were predicted to bind at the outer edge of the β-barrel structure with a ΔG values of −9.04 kcal/mol and −10.12 kcal/mol, respectively (Figure 3B,C, Appendix A). These compounds may therefore provide an effect on GBM cells to mimic the cellular mechanisms of T2A.

### 2.5. Functional Screen of MIOS Inhibitors to Reduced GBM Cell Proliferation

To investigate the bioactivity of potential MIOS inhibitors, 25 compounds with the greatest ∆G values for MIOS binding were analysed using GBM cell proliferation inhibition as a readout for compound efficacy, initially in GL261 cells [40] and then in U87-MG cells [41] (Figure 4 and Appendix A). A comparison of proliferation inhibition effects over 9 days involved treatment with T2A (Figure 4A) and two potential MIOS inhibitors (Mi3 and Mi10, Figure 4B,C). In this analysis, GL261 cells exposed to T2A significantly decreased cell proliferation by 17 and 19% at 5 and 10 µM, respectively (*p* < 0.05) (Figure 4D). Similarly, Mi3 treatment at 5 and 10 µM also significantly decreased cell proliferation by 37 and 57%, respectively (*p* < 0.01) (Figure 4E). This effect was not shown for all compounds. For example, Mi10 did not reduce cell proliferation (Figure 4F). Repeating this analysis with U87-MG cells, T2A treatment reduced cell proliferation by 54 and 58% (*p* < 0.0001) (Figure 4G), as did Mi3 by 25% at both concentrations (*p* < 0.0001) (Figure 4H), and Mi10 treatment provided no effect (Figure 4I). Mi1 treatment also reduced cell proliferation in GL261 cells (Appendix A). Thus, through the analysis of GBM cell proliferation, two compounds (Mi3 and Mi1) were shown to provide a significant inhibition of cell proliferation in GBM cells.

### 2.6. Validation of the Function of Mi3 in D. discoideum Dependent upon MIOS in Cell Proliferation Inhibition and Autophagy Induction

Mi3 specificity through the inhibition of MIOS was initially assessed using the *D. discoideum* mios^-^ mutant. In these experiments, wild-type and mios^-^ cells were treated with a range of Mi3 concentrations for 5 days and cell proliferation was measured. The treatment of wild-type cells with Mi3 reduced cell proliferation in a dose-dependent manner, with a significant reduction at 2.5 µM by 39% (*p* < 0.0022) that increased to 66% at 10 µM (*p* < 0.0022) (Figure 5A). In these experiments, T2A treatment at 10 µM also significantly reduced cell proliferation by 73% (*p* < 0.0022). The effect of Mi3 on cell proliferation was structurally specific since the related compound Mi10, also predicted to bind human MIOS (∆G −9.11), showed a reduction in inhibitory effect to 21%. In comparison, mios^-^ cell treatment with Mi3 provided no reduction in cell proliferation up to 10 µM and no reduction following T2A treatment at 10 µM (Figure 5B). Thus, the effect of Mi3 and T2A on cell proliferation was dependent upon MIOS activity, confirming this specific mechanism of action.

In addition to cell proliferation, the specificity of Mi3 on autophagy induction in *D. discoideum* was also investigated. In these experiments, wild-type and mios^-^ cells expressing ATG8-GFP were exposed to Mi3 (12 µM) for 24 h (Figure 5C). Autophagosome formation was visualised, showing that Mi3 increased the relative autophagosome number from 0.35 autophagosomes per μm^3^ in control-treated cells to 0.75 autophagosomes µm^3^ cell following Mi3 treatment. This increase was not observed in the mios^-^ cells (Figure 5D), where there was a decrease in the autophagosome number from 0.69 to 0.45 autophagosomes per mm^3^ cell. Thus, the effect of Mi3 on autophagy induction was dependent upon MIOS activity, confirming this specific mechanism of action.

### 2.7. Validation of the Role of Mi3 to Inhibit mTORC1 Activity in GBM Cells

To further establish the potential use of Mi3 in GBM treatment, it was imperative to confirm the effect of Mi3 on mTORC1 activity. In order to measure mTORC1 activity, p-4EBP1 was used as a read out, as it is a direct substrate of mTORC1 [24,27] (Figure 6A). T2A was used as a positive control because it has an established role in mTORC1 inhibition [7,19] (Figure 6B). This analysis showed that Mi3 treatment at 5 and 10 µM significantly decreased p4EBP1 levels by 51 and 35%, respectively (Figure 6C). The total 4EBP1 levels were also measured to determine whether this decrease was dependent upon mTORC1 activity or through the reduced transcription of the 4EBP1 protein (Figure 6D). These results showed that 10 µM of Mi3 significantly decreased total 4EBP1 levels by 35%. An analysis of the ratio of p4EBP1 to total 4EBP1 as a readout for mTORC1 activity showed that 5 µM Mi3 treatment significantly decreased mTORC1 activity (*p* < 0.001) (Figure 6E). T2A treatment also decreased mTORC1 activity, but Mi10 did not, which may explain the lack of inhibition of cell proliferation observed in GBM cells in response to this compound. Thus, Mi3 and T2A inhibited mTORC1 activity in GBM cells.

### 2.8. Validation of the Role of Mi3 to Induce Autophagy in GBM Cells

To investigate a role for Mi3 in autophagy induction, GL261 cells were exposed to Mi3 (2.5, 5 and 10 µM) for 3 days and autophagosomes were visualised and quantified using an autophagosome marker (MDC, monodansylcadaverine [42]), with T2A (10 µM) and rapamycin (750 nM) as positive controls (Figure 7A–F). In these experiments, the autophagosome number increased following treatment with T2A by 491% (*p* < 0.001), similar to the 359% increase shown following rapamycin treatment (*p* < 0.05), consistent with earlier studies [8,11,12,43] (Figure 7G). Following Mi3 treatment, the autophagosome number also increased in a dose-dependent manner from 326% at 5 µM (*p* < 0.05) up to 452% at 10 µM (*p* < 0.001). Thus, in GBM cells, Mi3 functions to induce autophagy, similar to what is seen for T2A treatment and in the *D. discoideum* model. 

## 3. Discussion

Autophagy is a highly conserved intracellular process required for the efficient degradation and recycling of cellular components [44,45], including misfolded proteins and damaged DNA that, when dysregulated, can contribute to cancer development [46]. T2A has been widely demonstrated to activate autophagy and inhibit cell proliferation through a range of molecular mechanisms including AMPK, MAPK, and PI3K signalling pathway regulation in various cancer models [8,11,12]. More recently, one study in *D. discoideum* proposed a distinct pathway of T2A through SESN to regulate mTORC1 [19] but did not provide evidence for a link to autophagy. Thus, this study employed *D. discoideum* to investigate this pathway of T2A in autophagy induction, since it is an established model to study autophagy [24,25,47,48], with proteins orthologous to mammalian models [49] and responses to pharmacological interventions comparable to mammalian and human cells. Thus, the mechanism of autophagy induction by T2A was investigated, focusing on the SESN to mTORC1 pathway, where mTORC1 has an established role in autophagy induction [50,51]. This analysis demonstrated that autophagy induction and mTORC1 inhibition by T2A in *D. discoideum* was dependent upon both SESN and a component of GATOR2, MIOS. This outcome proposed, for the first time, that MIOS may be a novel pharmaceutical target to reproduce the effect of T2A in cancer treatment. An mTORC1 inhibitory mechanism through targeting MIOS is potentially advantageous, since it may overcome changes in SESN regulation in cancers associated with p53 mutations [19,22]. This approach identified a MIOS-inhibitory compound, Mi3, that reproduced the effect of T2A on both *D. discoideum* and GBM cell proliferation and in autophagy induction.

Drug-resistant cancers are often associated with a loss of p53 activity [52,53], so it remains important to investigate novel anticancer treatments independent of p53 function. This was particularly important in relation to T2A anticancer activity, where the T2A-dependent reduction in GBM cell proliferation and tumour (spheroid) formation was found to be greatest in p53-positive GBM cells [19]. To address this consideration, a major component of the GATOR2 complex, MIOS, was targeted, as four MIOS subunits provide a major component of GATOR2 to regulate mTORC1 activity [36]. SESN binds to the GATOR2 complex, leading to the inhibition of GATOR1 (a negative regulator of Rag proteins) and subsequently mTORC1 [54]. The loss of GATOR2 components resulted in reduced mTORC1 activity [55], leading to increased autophagy [56,57]. Previous studies showed that T2A inhibited mTORC1 in a SESN-dependent manner [19]. The regulation of SESN expression has not been clearly defined, as some studies point to a p53-dependent [58] and independent mechanism [59]. As p53 is commonly dysregulated in 50% of cancer types [60,61,62], it provides a valid approach to target a protein further downstream than SESN to inhibit mTORC1 activity. Through the ablation of MIOS, the T2A-dependent inhibition of mTORC1 and autophagy induction was shown to be dependent upon MIOS activity, and these effects were reproduced with the novel MIOS inhibitor Mi3. To our knowledge, this was the first demonstration of targeting MIOS to develop a novel anti-cancer compound based upon the mechanism of T2A.

Developing novel cancer treatments requires a complex process, including target identification, screening protocols to identify novel compounds, and hit validation, and this process is costly and has a failure rate of 90% [63]. Thus, more robust and targeted approaches to identifying novel treatments are required. In this study, the MIOS protein was chosen as a target in the SESN to mTORC1 pathway, and novel compounds were identified that were predicted to bind strongly to the human MIOS protein. The screening protocol started with the in silico screening of compound databases against the β-propeller of MIOS, since this secondary structure is likely to be involved in the binding of MIOS into the GATOR2 structure [36]. From this screen, ∆G values were used to shortlist compounds, as thermodynamic characteristics provide an effective means of identifying active compounds [64]. The screening protocol continued with the functional activity analysis of compounds to reduce GBM cell proliferation, investigated in two GBM lines (GL261 and U87-MG), to identify a consistent effect of Mi3. These studies employed two cell lines, since GBM cell heterogeneity underlies variable responses to potential medicinal treatments [65,66] and the effects are also evident in many other cancer types [65,67,68]. The resulting compound validation was undertaken using two approaches. Firstly, the *D. discoideum* mios^-^ cells were used to confirm the specificity of Mi3 functioning through MIOS in cell proliferation reduction, mTOR inhibition, and autophagy induction, with a similar loss of potency towards T2A effects. Secondly, Mi3 was shown to inhibit mTORC1 and induced autophagy in GBM cells to again validate the mechanism of this compound. Therefore, this study outlined a protocol for cancer drug discovery, combining computational binding analysis, cancer cell culture analysis of selected compounds, and the validation of target specificity by demonstrating a loss of compound potency in the *D. discoideum* cell line.

In addition to developing T2A mimetics that function independently of p53 activity, the novel cancer treatment pipeline described here may overcome other limitations. Another considerable drawback to the widespread use of T2A as a medicinal treatment is poor bioavailability due to a high log *p* value and a poor water solubility [69]. These complications may be bypassed through the development of lipid nanocapsules [70] and biotinylated-lipid-bilayer-coated mesoporous silica nanoparticles [69]. However the identification and development of novel MIOS-targeting compound with improved chemical characteristics [71] will increase bioavailability and likely therapeutic efficacy in patient treatment. Thus, the further analysis of Mi3 bioavailability and the continued screening of novel inhibitors will likely identify compounds with increased plasma levels and potential brain exposure in GBM treatment.

## 4. Materials and Methods

### 4.1. Chemical Compounds

T2A was provided by AdooQ Bioscience (A10890) (Irvine, CA, USA). Potential MIOS inhibitors were purchased from Molport (Beacon, NY, USA) (Appendix A).

### 4.2. Cell Culture

Wild-type and mutant *D. discoideum* cells were grown in liquid HL5 media. GL261 [40] was grown in Dulbecco’s Modified Eagle Medium (DMEM, Gibco, Grand Island, NY, USA) supplemented with 10% Foetal Bovine Serum (FBS, PAN-Biotech, Aidenbach, Germany). Incubation was performed at 37 °C with 5% CO_2_. All grouped experiments were performed at the same time, providing direct comparisons to specific treatments. U87-MG is an established cell line that was cultured in EMEM (Gibco, NY, USA) supplemented with 10% FBS (Gibco, NY, USA), 5% non-essential amino acids (Gibco, NY, USA) and 5% sodium pyruvate (Gibco, NY, USA).

### 4.3. D. discoideum Cell Proliferation Assays

Cell proliferation was measured in liquid culture (HL5 media, Formedium^TM^, Norfolk, UK), maintained in the dark at 22 °C in 24-well plates (with 5000 cells in 500 μL HL5 media per well), and treated with varying concentrations of compounds at a constant solvent level (DMSO, 0.8%). Cells were incubated for seven days, and cell densities were determined from day 3 to day 7 and normalised to the solvent-only control.

### 4.4. Quantification of Cell Signalling by Western Blotting

*D. discoideum* cells were treated during proliferation for the indicated times with the control (solvent-only) or compounds of 12–25 µM in line with previous studies and publications in mammalian cancer cell lines [7,72,73] with all conditions containing 0.2% solvent. Protein samples were prepared by directly lysing 7.5 × 10^7^ cells/mL cells in 2 × Laemmli buffer (0.004% bromophenol blue, 10% 2-mercaptoethanol, 20% glycerol, 4% SDS, and 0.125 M Tris-HCl; Sigma, Sofia, Bulgaria) followed by boiling at 96 °C for 6.5 min. GBM human and mouse cells were treated during proliferation as above, where the final cell concentration was 1 × 10^6^ cells/mL. Protein samples were prepared using RIPA buffer and a protease cocktail inhibitor, stored in 6 × loading buffer, and boiled for 5 min at 100 °C. From 6 to 20 µL of each sample was separated by sodium dodecyl sulphate polyacrylamide gel electrophoresis (stacking gel: 5%; resolving gel: 10–15%). After proteins were transferred to the appropriate membrane (polyvinylidene difluoride or nitrocellulose, pore size: 0.2–0.4 µm; Thermo Fisher Scientific, Millipore, Amersham, UK), membranes were stained with Ponceau S dye and then blocked for 1 h with 5% bovine serum albumin (in TBS) or intercept (TBS) blocking buffer (LI-COR, Cambridge, UK). Membranes were incubated overnight at 4 °C with primary antibodies detecting phosphor (Thr37/46)-4EBP1 (1:1000, Cell Signalling Technology, 9459, Danvers, MA, USA) and 4EBP1 (Cell Signalling Technology, 9644S, USA). All primary antibodies were dissolved in 5% BSA (TBST) or intercept (TBS) blocking buffer containing 0.1% Tween 20 (Sigma, Bulgaria). As a loading control, either streptavidin Alexa Fluor 680 conjugate (1:5000, Invitrogen, S21378, Carlsbad, CA, USA) for methyl crotonyl-CoA carboxylase (MCCC1) [74], Alexa Fluor 690 conjugate (1:5000, LI-COR, 926-68070, UK) for β-actin (Sigma, A228, Macclesfield, UK), or Ponceau S-stained total protein was used. Membranes were washed with TBST and subsequently incubated for 1 h with IRDye800CW goat anti-rabbit IgG (1:10,000, LI-COR, UK) diluted in intercept (TBS) blocking buffer containing 0.1% Tween 20 and 0.01% SDS. After membranes were washed with TBST, protein levels were visualised using the Odyssey CLx imaging system (LI-COR, UK). Levels of the protein of interest were normalised to either the MCCC1, β-actin, or total protein (Ponceau S stain).

### 4.5. Generation of CRISPR Knockouts

Wild-type (Ax3) cells were transformed with the pTM1285 plasmid encoding the *mios* gene target sequences, designed to target exon 5 of the MIOS gene, and knockout were generated as previously described [24]. Target gene sequences were analysed to identify mutations (Appendix A). Primer sequences were MIOS oligo 1 fwd. GTAACTCAAACTAATCATCA, MIOS oligo 1 rev TGATGATTAGTTTGAGTTAC, MIOS oligo 2 fwd. AGTATTTCAACTACAACACC, MIOS oligo 2 rev GGTGTTGTAGTTGAAATACT, MIOS oligo 3 fwd. TTGGAGAATATATTATCAAG, and MIOS oligo 3 rev CTTGATAATATATTCTCCAA.

### 4.6. Autophagosome Formation Analysis (Atg8-GFP)

Cells were made to express GFP-Atg8 (as previously described [25,47]) and cultured to a density of 1.33 × 10^6^ cells/mL in 2 mL of HL5 media for control and treatment or KK2 for starvation conditions. Treatments included 12 µM of T2A or solvent DMSO control for 24 h. Starvation conditions involved the incubation of cells in KK2 for 4 h. Shaking was continued for 1 h before cells were removed (without additional pipetting). Cells were imaged live under KK2 1% agar gel after cells were resuspended in KK2. The time in KK2 and time under the agar were kept consistent between each repeat. Cells were imaged using an Olympus IX71 wide-field fluorescence microscope and captured with a MicroPix camera, model Elite 2 (Chichester, UK). For the measurement of Atg8-positive structures, cells were analysed using a MicroPix analysis tool with which the autophagosome number was normalised to cell size, which was measured across the largest diameter. Approximately 8 cells were analysed per photograph.

### 4.7. GBM Cell Proliferation Assays

GBM cells were seeded to a final concentration of 1 × 10^5^ cells and were treated with the compounds at a concentration of 0, 5 or 10 µM. Cells were incubated for 0, 3 and 6 days, and cell cytotoxicity was measured using sulforhodamine B assay (SRB) (Sigma, UK).

### 4.8. Computational Compound Screening

Human protein structures were sourced from the NCBI protein database (https://www.ncbi.nlm.nih.gov/protein/ (accessed on 3 February 2023)) and uploaded to the mcule database. The use of the hit identification function allowed us to screen with the mcule library of ~4 million compounds (https://mcule.com/hit-identification/ (accessed on 3 February 2023)). The top 50 compounds were individually docked using Swiss dock (http://www.swissdock.ch/ (accessed on 3 February 2023)). The top 25 compounds were sourced using Molport (https://www.molport.com/shop/index (accessed on 3 February 2023)).

### 4.9. GBM Autophagy Assay

GL261 cells were seeded onto coverslips at a density of 1 × 10^5^ cells per ml and after 24 h treated with either solvent alone, T2A (25 µM) or Mi3 at 2.5, 5 or 10 µM for 3 days. Rapamycin was used as a control (750 nM, 16 h). Autophagy was detected using the Autophagy kit (Abcam, ab139484, Cambridge, UK) following the manufacturer’s instructions. Briefly, a dual detection fluid including an autophagy-vacuole-specific FITC antibody (ab139484) and a nuclear counterstain (Hoeschst 33342) was incubated with the treated cells, followed by a fixation step with 4% paraformaldehyde. Cells were mounted and sealed onto microscope slides, and fields of view were imaged using a x60 water objective on a LSM 900 Zeiss confocal microscope (Ziess, Jean, Germany). Cells were analysed for autophagosome number by counting autophagosomes per cell in 5–8 fields of view per experiment and calculating averages.

### 4.10. Statistics

The data are represented as mean ± SEM. Statistical significances between two groups were analysed using a two-tailed Mann–Whitney test (GraphPad PRISM, version 10).

## 5. Conclusions

T2A has been shown to induce autophagy in cancer models [8,11,12] without a clear molecular mechanism. Using *D. discoideum*, this study showed that T2A treatment induced autophagy and inhibited mTORC1 through a mechanism dependent upon SESN and MIOS proteins. This demonstrated a molecular pathway for T2A through SESN and GATOR2 signalling to mTORC1 and autophagy. Since SESN regulation is dependent upon p53, where this protein is often at reduced level or mutated in refractory cancers, the study further developed an approach for novel anticancer compound discovery, targeting MIOS to reproduce the effect of T2A and thus leading to the identification and validation of Mi3. The treatment of GBM cells with the novel MIOS inhibitor reproduced the T2A-dependent inhibition of GBM cell proliferation and mTORC1 inhibition and induced autophagy. This paper therefore provides a streamlined drug discovery approach for new GBM treatments.

## Figures and Tables

**Figure 1 ijms-25-06586-f001:**
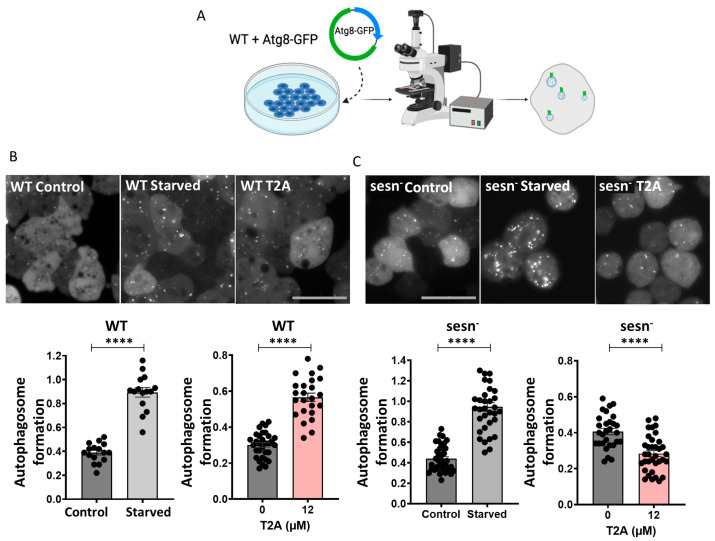
T2A treatment triggered autophagy induction in *D. discoideum* dependent upon SESN. (**A**). Quantification of autophagy in *D. discoideum* requires the expression of ATG8-GFP to visualise fluorescent autophagosomes. (**B**). In wild-type cells, comparing solvent-only (DMSO) conditions with starvation conditions (4 h in buffer) and T2A treatment (12 µM for 24 h in media) indicated increased autophagosome formation, as quantified per cell normalised to the cell size (per µm^3^). (**C**). In sesn^-^ cells, the number of autophagosomes increased under starvation conditions (4 h in salt buffer) but decreased following T2A treatment (12 µM for 24 h in media), as quantified per cell normalised to the cell size (per µm^3^). Data are derived from at least triplicate experiments, and all data are shown as mean ± SEM; **** *p* ≤ 0.0001 (two-tailed unpaired Mann–Whitney test and non-parametric data). Scale bar—25 µm.

**Figure 2 ijms-25-06586-f002:**
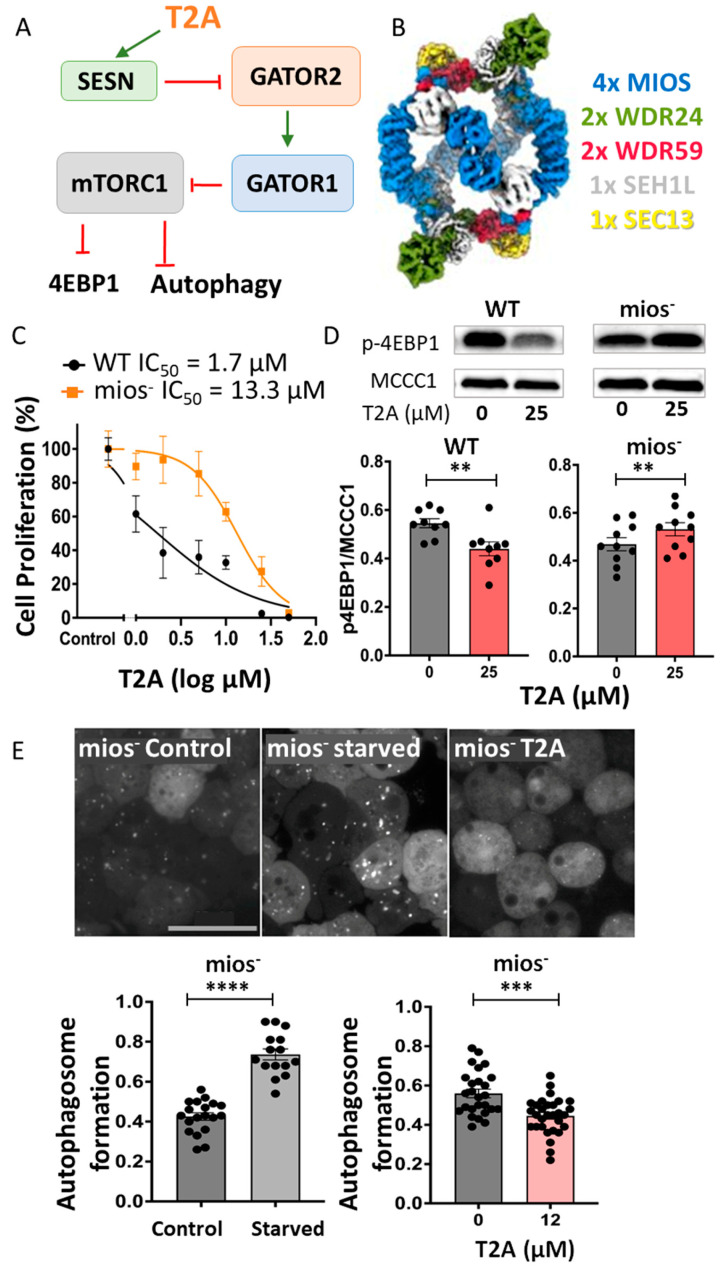
T2A regulation of mTORC1 activity, cell proliferation and autophagy induction is dependent upon MIOS. (**A**) The pathway from SESN to mTORC1 may progress through the GATOR function, where T2A treatment inhibits mTORC1 (measured by reduced 4EBP1 phosphorylation), and may control autophagy induction (red connector indicates inhibition). (**B**) The GATOR2 complex, shown as a 3D structure, includes MIOS (blue), WDR24 (green), WDR59 (pink), and accessory proteins of SEH1L and SEC13 (in white and yellow, respectively). Image from [36] with permission. (**C**) The ablation of MIOS (in mios^-^) reduced cell proliferation sensitivity to T2A, reflected in an increased IC_50_ value in the mutant, and (**D**) blocked the effect of T2A (25 µM for 24 h) in reducing mTORC1 activity, as measured by 4EBP1 phosphorylation. (**E**) The ablation of MIOS (in mios^-^) had no effect on autophagosome induction in starvation conditions but inhibited T2A-dependent autophagosome induction, as quantified per cell normalised to the cell size (per µm^3^). All data are shown as mean ± SEM; ** *p* ≤ 0.01, *** *p* ≤ 0.001, **** *p* ≤ 0.0001 (two-tailed unpaired Mann–Whitney test and non-parametric data). Scale bar—25 µm.

**Figure 3 ijms-25-06586-f003:**
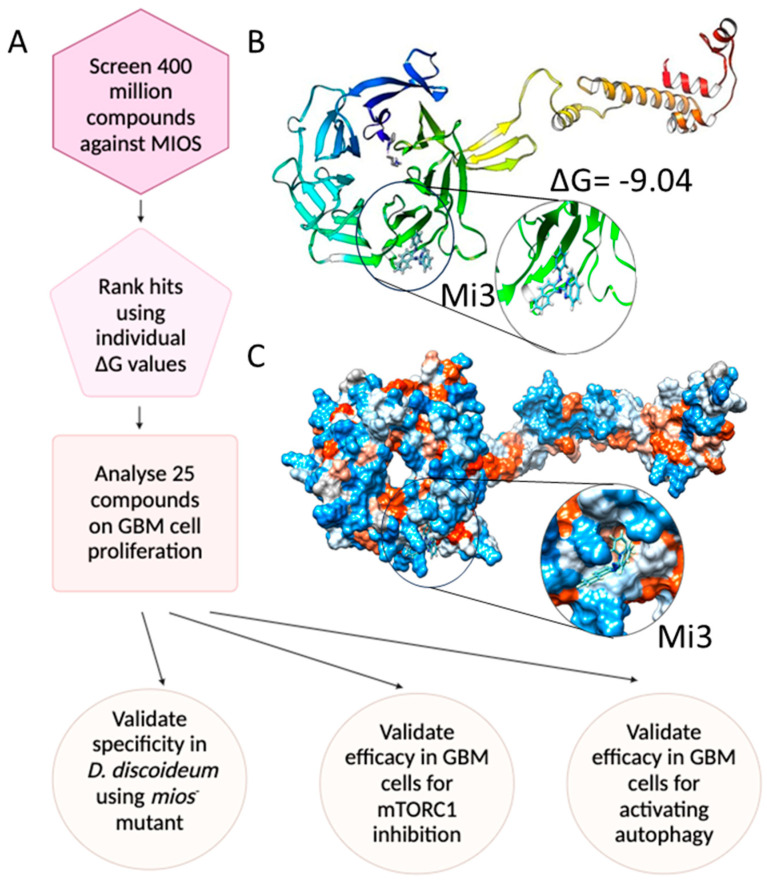
Workflow diagram for compound library screens to identify novel MIOS inhibitors. (**A**) Workflow illustration of the in silico investigation used to identify novel MIOS inhibitors by docking analysis and analysis of GBM cell line proliferation, followed by specificity analysis in *D. discoideum* mios^-^ cells to show loss of bioactivity for mTORC1 inhibitory activity and autophagy induction in GBM cells. (**B**) Ribbon diagram representation of the human MIOS protein showing predicted binding of Mi3 to the outer edge of the β-barrel structure. (**C**) Space filling hydrophobicity representation of the human MIOS protein indicating the predicted Mi3-binding pocket.

**Figure 4 ijms-25-06586-f004:**
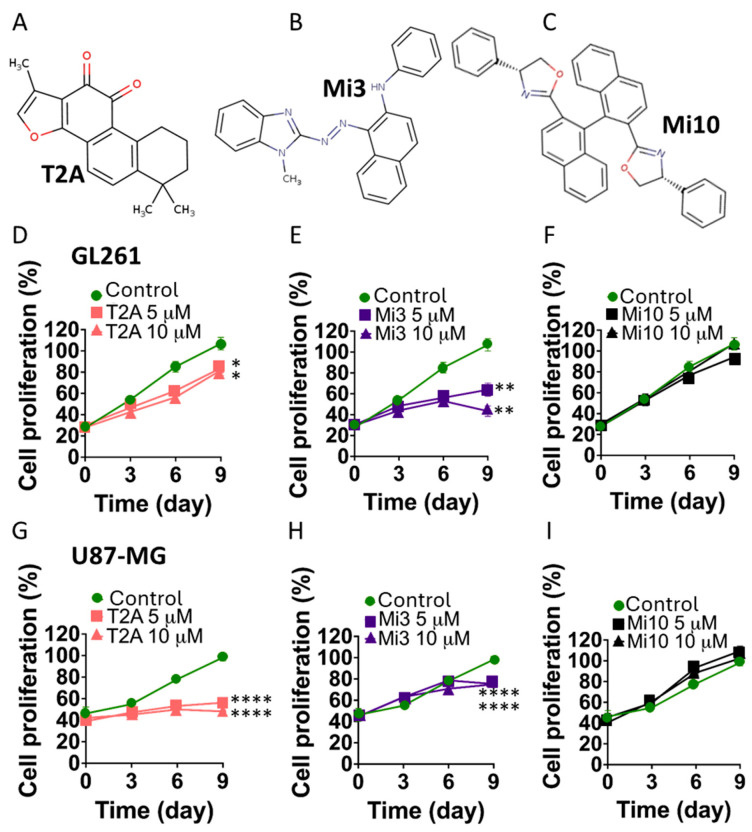
T2A and a novel MIOS inhibitor reduce cell proliferation in two GBM cell lines. GL261 and U87-MG cell lines were treated with 5 or 10 µM of T2A, the novel MIOS inhibitors (Mi3 or Mi10), or a solvent-only control and cultured for 9 days. All results are shown normalised to control treatment on day 9. (**A**–**C**) Structures of the compounds are shown. (**D**–**F**). GL261 cells treated with T2A (pink) and Mi3 (purple) showed reduced cell proliferation at both 5 and 10 µM, whereas treatment with Mi10 (black) had no effect. (**G**–**I**). U87 cells treated with T2A and Mi3 showed reduced cell proliferation at both 5 and 10 µM, whereas treatment with Mi10 had no effect. All data are shown as mean ± SEM; * *p* ≤ 0.05, ** *p* ≤ 0.01, **** *p* ≤ 0.0001 (two-tailed unpaired Mann–Whitney test and non-parametric data).

**Figure 5 ijms-25-06586-f005:**
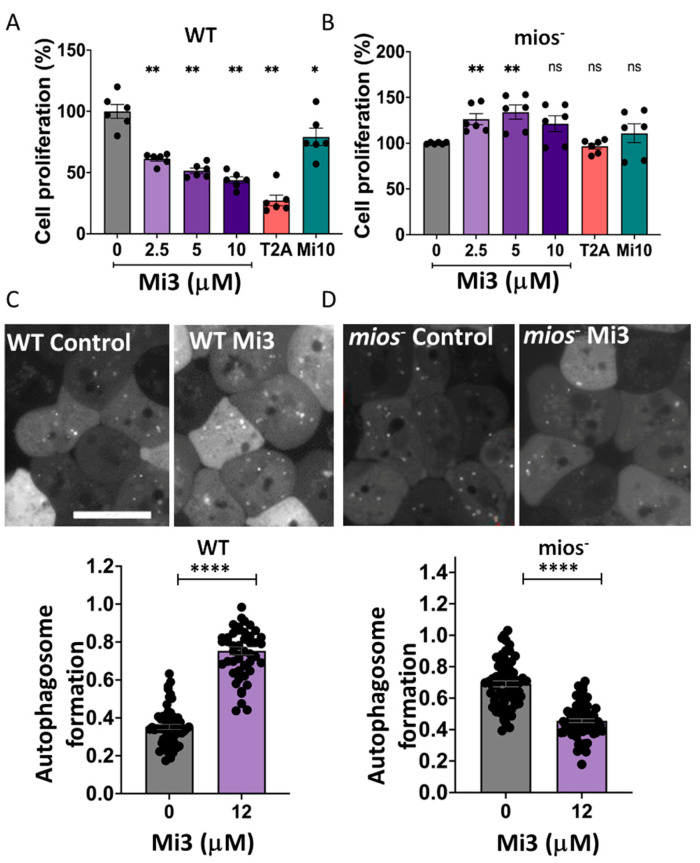
T2A and Mi3 inhibit cell proliferation and activate autophagy dependent upon MIOS in *D. discoideum*. (**A**) *D. discoideum* wild-type cells treated with Mi3 (0–10 µM) showed a dose-dependent reduction in proliferation, consistent with T2A treatment (10 µM), whereas treatment with a related structure, Mi10 (10 µM), showed a reduced inhibition of cell proliferation. All cells were treated for 5 days, and proliferation normalised to solvent-only conditions. (**B**) In contrast, in mios^-^ cells, these inhibitory effects were blocked. (**C**) In autophagy induction, wild-type cells treated with Mi3 induced autophagosome induction, and (**D**) this effect was lost in mios^-^ cells, as quantified per cell normalised to the cell size (per µm^3^). All data are shown as mean ± SEM; * *p* ≤ 0.05, ** *p* ≤ 0.01, **** *p* ≤ 0.0001, ns—not significant (two-tailed unpaired Mann–Whitney test and non-parametric data). Scale bar—25 µm.

**Figure 6 ijms-25-06586-f006:**
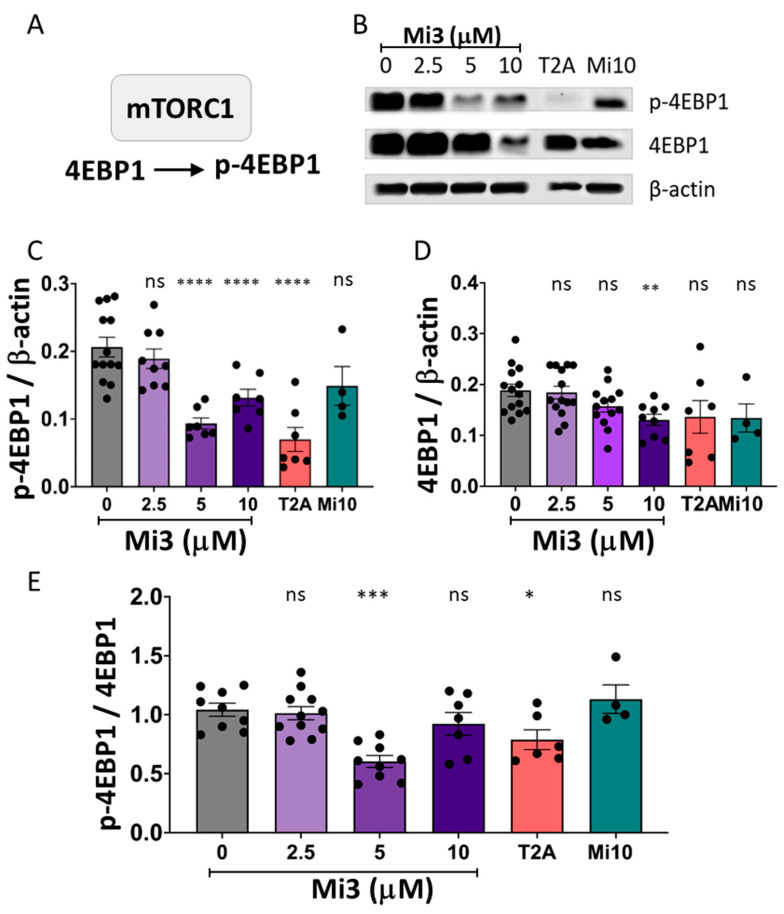
Mi3 inhibition of mTORC1 activity in GBM cells. **(A**) mTORC1 activity can be quantified through analysis of phosphorylated 4EBP1 (p4EBP1) and total 4EBP1. (**B**) Western blot images from GL261 cells treated with Mi3 (2.5–10 µM), and T2A or Mi10 (10 µM) for 6 days. (**C**,**D**) Quantification of p-4EBP1 and total 4EBP1 following treatment indicates a reduction in p-4EBP1 levels by Mi3 and T2A and a reduction in 4EBP1 by Mi3 (10 µM). (**E**) The ratio of p-4EBP1 to 4EBP1 shows significant inhibition of mTORC1 by Mi3 (5 µM) and T2A (10 µM) that was not shown by Mi10. All data are shown as mean ± SEM; ns—not significant, * *p* ≤ 0.05, ** *p* ≤ 0.01, *** *p* ≤ 0.001, **** *p* ≤ 0.0001 (two-tailed unpaired Mann–Whitney test and non-parametric data).

**Figure 7 ijms-25-06586-f007:**
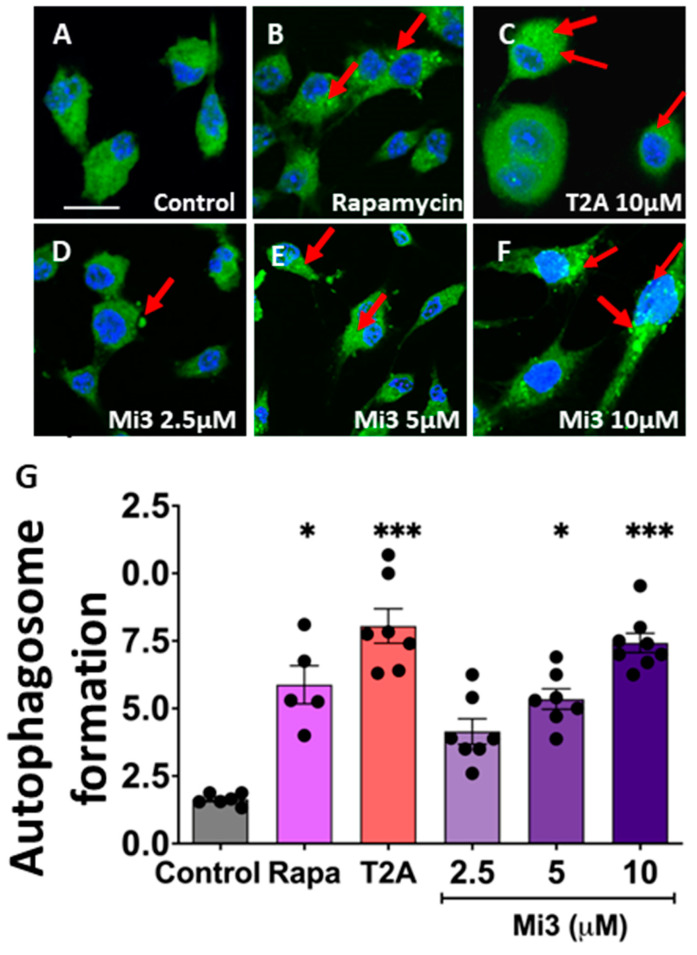
Mi3 induction of autophagy in GBM cells. (**A**–**F**). Representative images of GL261 cells under control (solvent-only) conditions or following treatment with rapamycin (750 nM; 16 h), T2A (10 µM), and Mi3 (as indicated) for 3 days; cells were stained for autophagosomes (examples indicated by red arrows) using an autophagy-vacuole-specific FITC antibody (green) and a nuclear counterstain (blue) (Hoeschst 33342). (**G**). Quantification of autophagosome formation (mean autophagosome number per cell) following treatment, with data derived from independent experiments with 5–7 fields of view analysed per treatment. All data are shown as mean ± SEM, * *p* ≤ 0.05, *** *p* ≤ 0.001 (Kruskal–Wallis with Dunn’s multiple comparison test). Scale bar—10 µm.

## Data Availability

All data are provided in the manuscript.

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
