# Peer review of "Developing a Tanshinone IIA Memetic by Targeting MIOS to Regulate mTORC1 and Autophagy in Glioblastoma"

_ijms, 2024, doi:10.3390/ijms25126586_

Round 1

Reviewer 1 Report

Comments and Suggestions for Authors

The manuscript entitled “Developing a Tanshinone IIA Memetic through Targeting MIOS Regulates mTORC1 and Autophagy in Glioblastoma” is a good piece of research, focusing on the investigation of the mechanism of action of T2A in the induction of autophagy and to develop a process for developing T2A mimetics using the model system Dictyostelium discoideum and GBM cell lines. The manuscript needs revision on the bases of the comments given below. 

1. Use passive language in the entire manuscript

2. Keywords to be written alphabetically

3. Write “Danshen (Salvia miltiorrhiza Bunge)” instead of “Danshen, Salvia miltiorrhiza Bunge” (line 29-

     30 and in the rest of the manuscript)

4. Reason for selecting Danshen (Salvia miltiorrhiza Bunge) as compared to other medicinal plants containing the same bioactive compound?

5. Extract selection (ethanolic, methanolic, water, or anyother organic solvent?

6. Background of the study is either based on the preliminary tests of the extracts or the literature review?

7. There are many model organisms such as Saccharomyces cerevisae with numerous conserved genes and orthologous proteins associated with human diseases: why was the preference given to D. discoideum?

8. The last paragraph of the Introduction should have the crux of the study; it may have the objectives or the main purpose of the study. The current manuscript contains the results in that paragraph which need to be omitted and this paragraph is to be rephrased.

Comments on the Quality of English Language

The paper is well written except few typos errors/grammatical mistakes. The authors may read the manuscript carefully again and can address the issue easily. 

Author Response

Reviewer 1

The manuscript … is a good piece of research, focusing on the investigation of the mechanism of action of T2A in the induction of autophagy and to develop a process for developing T2A mimetics using the model system Dictyostelium discoideum and GBM cell lines.

We thank the reviewer for these comments and suggestions. It was very nice to see the approval of all the general sections (background, research design, methods, results, conclusion, references).

We note that minor editing of the language in the manuscript has been requested and have complete this in the resubmission.

Regarding specific comments and suggestions, we address these individually in the below.

  1. Use passive language in the entire manuscript

We have been through the manuscript to ensure that passive language is used throughout.

  1. Keywords to be written alphabetically

Agreed and completed.

  1. Write “Danshen (Salvia miltiorrhiza Bunge)” instead of “Danshen, Salvia miltiorrhiza Bunge” (line 29-30 and in the rest of the manuscript)

Agreed and completed.

  1. Reason for selecting Danshen (Salvia miltiorrhiza Bunge) as compared to other medicinal plants containing the same bioactive compound?

T2A is found in Danshen, and numerous papers have published the potential for T2A in cancer treatment. We feel it is important to provide at least one origin of T2A and have chosen one of the mostly commonly known medicinal plant to use as an example. It is not clear to us why the reviewer wants us to evidence other sources of T2A.

  1. Extract selection (ethanolic, methanolic, water, or any other organic solvent?

The source of T2A is provided in the chemical compounds section of the manuscript, and no extraction was undertaken in the manuscript. Details of the compound are provided by the supplier.

  1. Background of the study is either based on the preliminary tests of the extracts or the literature review?

It is unclear to me what the reviewer refers to here. The background of the study is detailed in the manuscript introduction, referring to 31 published papers. Furthermore, we employed a purified, commercially sourced product (>98% purity), not extracts as mentioned in the comment.

  1. There are many model organisms such as Saccharomyces cerevisae with numerous conserved genes and orthologous proteins associated with human diseases: why was the preference given to D. discoideum?

It is unclear to us what the reviewer is requesting here. Our research group employs Dictyostelium as a model for research, which we have used for many studies. So yes, the reviewer’s comment is correct, and there are many other models, and cell lines that can be used. Paragraph 3 in the introduction provide 184 words and 9 publications indicating why this model was employed.

  1. The last paragraph of the Introduction should have the crux of the study; it may have the objectives or the main purpose of the study. The current manuscript contains the results in that paragraph which need to be omitted and this paragraph is to be rephrased.

We thank the reviewer for this suggestion. As requested, we have reviewed this paragraph to remove the results in this paragraph and instead list the objectives in the study.

Reviewer 2 Report

Comments and Suggestions for Authors

Introduction:

1. Tanshinone IIA (T2A) is a bioactive compound derived from the root extract of Dans- 29 hen, Salvia miltiorrhiza Bunge. Please mention the family name of the species.

2. T2A has been demonstrated to function as a cancer treat- 30 ment through a range of pathways: the author should mention the names of various pathways.

Results: The author should emphasize more on the results. The results should be more explanatory.

Discussion: This section should be more elaborative. Moreover, figure no should be properly cited according to the text.

The author should add a conclusion section in the manuscript.

Comments on the Quality of English Language

The manuscript should be thoroughly revised to correct grammatical and punctuation errors. 

Author Response

Reviewer 2

We note that the reviewer states that our study provides a suitable study design, but needs improvement in  background, methods, results, conclusion, reference. We have these points below and in the resubmission of the manuscript with track changes.

We also note the reviewer indicates that minor editing of the language in the manuscript has been requested and we have completed this in the resubmission.

Regarding specific comments and suggestions, we address these individually in the below.

  1. Introduction. Tanshinone IIA (T2A) is a bioactive compound derived from the root extract of Danshen, Salvia miltiorrhiza Bunge. Please mention the family name of the species.

We thank the reviewer for this and have included this information.

  1. Introduction. T2A has been demonstrated to function as a cancer treatment through a range of pathways: the author should mention the names of various pathways.

We thank the reviewer for this suggestion and have reorganised this paragraph to address this point.

  1. Results. The author should emphasize more on the results. The results should be more explanatory.

We have made a range of changes throughout the results section to better explain the results.

  1. Discussion. This section should be more elaborative. Moreover, figure no should be properly cited according to the text.

We have updated the discussion to provide more attention to completeness and detail and removed figure citations.

  1. The author should add a conclusion section in the manuscript.

We note that the instructions for authors for the journal state that conclusions are optional (‘This section is not mandatory but can be added to the manuscript if the discussion is unusually long or complex’) and we do not think that the discussion is long or complex. However, to comply with the reviewer’s request, we have included this section.

  1. The manuscript should be thoroughly revised to correct grammatical and punctuation errors.

We thank the reviewer for this suggestion and have reviewed the manuscript thoroughly to ensure that we have corrected any grammatical or punctuation errors.

Reviewer 3 Report

Comments and Suggestions for Authors

After reviewing the manuscript titled: Developing a Tanshinone IIA Memetic through Targeting 2 MIOS Regulates mTORC1 and Autophagy in Glioblastoma By Sonia Shinhmar et al., I noticed that the paper is generally well written and structured. The author justified the research challenge and questions, identified gaps, shortcomings, and limitations in existing research, gave context to their manuscript, demonstrated a thorough and current understanding of the peer-reviewed literature relating to the topic, and provided the most pertinent literature related to the challenge. Moreover, the manuscript article is technically sound, and the data support the conclusions.

I believe the manuscript is suitable for publication in IJMS after the authors could address the following few comments and questions.

Comments to the Author

Minor Revisions

1.     Abstract:

 ·        More clarity on the rationale of the study is still required.

2.     Abbreviations: should be defined at 1st mention; some of the definitions were not written

o   Line 15: Please add the full name beside GATOR2 and MIOS.

o   Line 37: Please add the full name beside AMPK, MAPK, and PI3K. 

3.     Introduction:

 ·        Line 32: Many references were cited in this manuscript; the authors cited 9 papers with the same objectives,  and I think this was too much. Please explain why.

4.     Materials and Methods:

 ·        Clarify why you select to treat cells with T2A (12 µM, 24 hours). Clarify why you select this dose. Where are your references? Is this considered an acceptable dose?

 ·        Why did you not try different doses or different hours to demonstrate your hypothesis?

 ·        Authors should add the make and country of all chemicals and instruments used in the study.

 ·        Line 393: the authors wrote with 10% Foetal Bovine Serum 393 (FBS, PAN-Biotech). Incubation refers to 37°C with 5% CO2, which was already written in lines 391 and 392; please revise.

5.     Results:

             ·       The resolution of Figures 2E, 5C, and 7(A-F) is very low; they need to be improved, please modify them.

6.     The conclusion seems short and comprehensive but needs to be separated from the discussion with more details. The authors should rephrase their conclusion to align with their findings.

Best regards

Author Response

Reviewer 3

‘I noticed that the paper is generally well written and structured. The author justified the research challenge and questions, identified gaps, shortcomings, and limitations in existing research, gave context to their manuscript, demonstrated a thorough and current understanding of the peer-reviewed literature relating to the topic, and provided the most pertinent literature related to the challenge. Moreover, the manuscript article is technically sound, and the data support the conclusions.

I believe the manuscript is suitable for publication in IJMS after the authors could address the following few comments and questions’.

We thank the reviewer for these comments and suggestions. It was very nice to see the approval of the general section (background, research design, methods, results, references).

We note the indication that minor editing of the conclusion has been requested that has been complete in the resubmission.

We also note that the English language used in the manuscript reviewed as ‘fine, with no issues detected’.

Regarding specific comments and suggestions, we address these individually in the below.

  1. Abstract: More clarity on the rationale of the study is still required

We thank the reviewer for this and have modified the abstract to highlight the rationale: ‘Here we employed the model system Dictyostelium discoideum and GBM cell lines to investigate the cellular role of T2A, focusing on the regulation of SESN to function through a GATOR2 component MIOS, to inhibit mTORC1 and activate autophagy’.

  1. Abbreviations: should be defined at 1st mention; some of the definitions were not written (GATOR2 MIOS, AMPK, MAPK, and PI3K).

Although we feel that defining these commonly used abbreviations will reduce the clarity of the text, we have followed the reviewer’s suggestion and included these definitions and others (e.g. 4EBP1 etc).

  1. Introduction: Line 32: Many references were cited in this manuscript; the authors cited 9 papers with the same objectives, and I think this was too much. Please explain why.

We included these to highlight the depth of research in the area, but we agree with the reviewer’s comments here and have reduced the number of citations.

4a.  Materials and Methods: Clarify why you select to treat cells with T2A (12 µM, 24 hours). Clarify why you select this dose. Where are your references? Is this considered an acceptable dose?

We thank the reviewer for these points. We clarify these conditions by citing a recently published paper using these conditions in this model to regulate mTORC1 activity. Additionally, we use 12 µM T2A in these experiments as mammalian cell studies have shown to use a range of T2A concentrations of 10-50 µM in LNCaP prostate cancer cells, (doi: 10.1248/bpb.33.1828) and upto 80 µM in PC-3 cells (doi: 10.1002/ptr.5015), as well as 1-15 µM used in bladder cancer 5637 cells (doi:10.3390/ijms150915622). Note: There is one study in GBM cells where they used 7 µM to inhibit mTORC1 (https://doi.org/10.3892/or.2014.3293). The concentrations used were guided by both previous literature as well as our own previous experiments.

4b.  Materials and Methods: Why did you not try different doses or different hours to demonstrate your hypothesis?

The hypothesis of this part of the study was that T2A treatment would trigger the induction of autophagy in this model, and that this induction was dependent upon the presence of sestrin – not to determine a dose response. Our data confirms this hypothesis. We have modified the last paragraph of the introduction to clarify the goals of this study (as requested by Reviewer 1).

As shown in the analysis of this data (Fig. 1B and C), there is a distribution in the number of autophagosome formed per cell, making dose curve analysis of limited value.

4c.  Materials and Methods: Authors should add the make and country of all chemicals and instruments used in the study.

We thank the reviewer for this comment and have added country and company names to the chemicals and instruments used in this study whenever possible.

4d.  Materials and Methods: Line 393: the authors wrote with 10% Foetal Bovine Serum (FBS, PAN-Biotech). Incubation refers to 37°C with 5% CO2, which was already written in lines and 392; please revise.

We thank the reviewer for this and have corrected this error.

  1. Results: The resolution of Figures 2E, 5C, and 7(A-F) is very low; they need to be improved, please modify them.

We thank the reviewer for these comments. This may be due to the online submission software. However, to address these concerns, we have included 300 dpi large individual figures in the resubmission.

  1. The conclusion seems short and comprehensive but needs to be separated from the discussion with more details. The authors should rephrase their conclusion to align with their findings.

We note that the instructions for authors for the journal state that conclusions are optional (‘This section is not mandatory but can be added to the manuscript if the discussion is unusually long or complex’), and do not think the discussion is overlong or complex. However, to comply with the reviewer’s request, we have included this section and aligned the text to better reflect our findings.

Reviewer 4 Report

Comments and Suggestions for Authors

This research paper by Shinhmar et al. studies the "Developing a Tanshinone IIA Memetic through Targeting 2 MIOS Regulates mTORC1 and Autophagy in Glioblastoma". This work is a good contribution to the field and could be published in the International Journal of Molecular Sciences after major revision as mentioned below:

1.     Overall, the manuscript demonstrates a commendable grasp of English. The writing is clear and well-structured, facilitating easy comprehension for the reader. However, there are a few areas where improvements could enhance the overall quality of the manuscript. Like:

2.     Page 10, line 281, is Mi3 (10 10 µM) correct in the caption of Figure 6?

3.     There is no scale bar in Figure 7 (A-F). Please show the scale bar. Which dyes did you use for the green and blue staining?

4.     How does Tanshinone IIA (T2A) induce autophagy in glioblastoma multiforme (GBM) cells, and what is the role of the SESN2-MIOS-mTORC1 pathway in mediating this effect?

5.     Additional information needed on the techniques, such as virtual screening, molecular docking, or biochemical assays, would provide insights into how the authors identified potential T2A mimetics targeting MIOS.

6.     This work mentions that the Mi3 compound was identified through computational screening to mimic the mechanism of Tanshinone IIA (T2A) by binding to MIOS, a component of the GATOR2 complex. However, the search results do not provide mechanistic insights into how Mi3 specifically induces autophagy in GBM cells, and the molecular pathways involved.

7.     Figure 6 C is not quite dense and may be difficult for readers to interpret the statistics briefly. Redraw this figure. Figure 6 A also discusses the quantification of mTORC1 activity by measuring the phosphorylation of 4EBP1 to p—4EBP1. However, this schematic does not show it; redraw this part to show it in detail.

8.     The introduction part should be revised. Some recent literature in this field should be cited in the introduction parts. See for instance:

https://www.sciencedirect.com/science/article/pii/S0169433223002507

https://onlinelibrary.wiley.com/doi/abs/10.1002/adfm.202315040

Comments on the Quality of English Language

Moderate editing of the English language is required.

Author Response

Reviewer 4

‘This work is a good contribution to the field and could be published in the International Journal of Molecular Sciences after major revision as mentioned below’.

We thank the reviewer for these comments and suggestions. It was nice to see the approval of the methods section, with requested improvement of the background/references and research design requested.

We note the indication that moderate editing of the English language used in the manuscript is fine, with no issues detected.

Regarding specific comments and suggestions, we address these individually in the below.

  1. Overall, the manuscript demonstrates a commendable grasp of English. The writing is clear and well-structured, facilitating easy comprehension for the reader.

We thank the reviewer for this and have modified the last paragraph of the introduction to correct this.

  1. However, there are a few areas where improvements could enhance the overall quality of the manuscript. Like Page 10, line 281, is Mi3 (10 10 µM) correct in the caption of Figure 6?

We thank the reviewer for this and have corrected the error.

  1. There is no scale bar in Figure 7 (A-F). Please show the scale bar. Which dyes did you use for the green and blue staining?

We thank the reviewer for this comment. The size bar (in A) was too grey to easily see, so we have corrected this. Dyes used in this assay are described in the methods section, but we have updated the legend to improve clarity.

  1. How does Tanshinone IIA (T2A) induce autophagy in glioblastoma multiforme (GBM) cells, and what is the role of the SESN2-MIOS-mTORC1 pathway in mediating this effect?

We have shown here that in Dictyostelium, T2A induces autophagy dependent upon SESN-MIOS-mTORC1 and have identified a novel compound to specifically inhibit MIOS and trigger autophagy. We have also shown that the same compound inhibits mTORC1 and induces autophagy in GBM cells. Autophagy induction has been demonstrated through regulation of mTORC1 activity (doi: 10.1042/EBC20170027; doi: 10.3390/biom7030052), and thus the link between mTROC1 and autophagy is not the focus of this study.

We include text in the first paragraph of the discussion highlighting this established pathway.

  1. Additional information needed on the techniques, such as virtual screening, molecular docking, or biochemical assays, would provide insights into how the authors identified potential T2A mimetics targeting MIOS.

A general description of the screening experiments is provided in the results section (2.4). A technical description of these processes is provided in the Computational compound screening section of the methods. It is unclear from the reviewer’s text what further information is needed.

  1. This work mentions that the Mi3 compound was identified through computational screening to mimic the mechanism of Tanshinone IIA (T2A) by binding to MIOS, a component of the GATOR2 complex. However, the search results do not provide mechanistic insights into how Mi3 specifically induces autophagy in GBM cells, and the molecular pathways involved.

We thank the reviewer for these comments but respectfully disagree. Mi3 was identified from over 400 million compounds as a potential inhibitor of MIOS. It was then validated in this mechanism through mimicking the effect of T2A on reducing proliferation of two GBM lines, and similarly reducing proliferation of Dictyostelium where this effect was lost in the knockout MIOS cell line, indicating specificity of action. The molecular pathway of Sestrin to GATOR2 (including MIOS) to mTORC1 is well established, and a role for mTORC1 in autophagy is also well established (doi: 10.1042/EBC20170027; doi: 10.3390/biom7030052).

  1. Figure 6 C is not quite dense and may be difficult for readers to interpret the statistics briefly. Redraw this figure. Figure 6 A also discusses the quantification of mTORC1 activity by measuring the phosphorylation of 4EBP1 to p—4EBP1. However, this schematic does not show it; redraw this part to show it in detail.

We thank the reviewer for these comments and have updated the figure as requested. 

  1. The introduction part should be revised. Some recent literature in this field should be cited in the introduction parts. See for instance: https://www.sciencedirect.com/science/article/pii/S0169433223002507 https://onlinelibrary.wiley.com/doi/abs/10.1002/adfm.202315040

Unless the links provided here are incorrect, these papers have no apparent relation to the manuscript, and provide no mention of T2A, tanshinone, mios, autophagy, GATOR2, glioblastoma, Sestrin or Dictyostelium. We therefore cannot see a mechanism for citing these in the introduction. However, both suggested manuscripts have common authors (Alireza Hassani Najafabadi and Neda Mohaghaegh), and neither author appears to have published on ‘tanshinone’, ‘mios’ or ‘sestrin’ (from pubmed searching for these names and terms).

Round 2

Reviewer 4 Report

Comments and Suggestions for Authors

Accept.